# 3BTRON: A Blood-Brain Barrier Recognition Network

Nan Fletcher-Lloyd [1,2] ✉, Isabel Bravo-Ferrer[3,4,8], Katrine Gaasdal-Bech[3,4,5], Blanca Díaz Castro [3,4,5,9] & Payam Barnaghi [1,2,6,7,9]

The blood-brain barrier (BBB) plays a crucial role in maintaining brain homeostasis. During ageing, the BBB undergoes structural alterations. Electron microscopy (EM) is the gold standard for studying the structural alterations of the brain vasculature. However, analysis of EM images is time-intensive and can be prone to selection bias, limiting our understanding of the structural effect of ageing on the BBB. Here, we introduce 3BTRON, a deep learning framework for the automated analysis of electron microscopy images of the BBB. Using age as a readout, we trained and validated our model on a unique dataset ($n = 359$). We show that the proposed model could confidently identify the BBB of aged mouse brains from young mouse brains across three different brain regions, achieving a sensitivity of 77.8% and specificity of 80.0% post-stratification when predicting on unseen data. Additionally, feature importance methods revealed the spatial features of each image that contributed most to the predictions. These findings demonstrate a new data-driven approach to analysing age-related changes in the architecture of the BBB.

The blood vessels of the brain, unlike the vasculature in other organs, form a unique structure that tightly controls the exchange of substances between the blood and the brain parenchyma, thereby maintaining brain homeostasis. This structure is known as the blood-brain barrier (BBB). The BBB is comprised of the brain endothelial cells, which form the blood vessel wall and are tightly sealed by highly specialised intercellular junctions (tight junctions), supported by mural cells (smooth muscle cells and pericytes), astrocyte processes (endfeet), and the extracellular matrix (basement membrane) that surrounds them[1]. Ageing is accompanied by changes in the structure and function of the BBB[2,3]. Among these age-related changes are an increased BBB permeability to exogenous dyes[4,5] and increased transcytosis and molecular remodelling[6]. At the subcellular (ultrastructural) level, several studies have revealed changes in specific parameters such as tight junction tortuosity or basement membrane thickness[7,8]. However, the labour-intensive nature of analysing BBB ultrastructure images has limited a detailed understanding of the structural effect of ageing on the BBB and how this may contribute to BBB dysfunction. As ageing is considered a major risk factor for most neurodegenerative diseases[9], and vascular dysfunction is an early hallmark of diseases that lead to dementia[10,11], there is an unmet need for tools that allow for rapid, large-scale analysis to gain deeper insights into the impact of ageing on this crucial functional barrier.

While magnetic resolution imaging (MRI) studies have been able to uncover larger-scale changes in brain vasculature due to age[12-14], the spatial resolution of MRI, in the order of millimetres, makes the technique unsuitable to capture changes occurring at the cellular and sub-cellular level. Similarly, light microscopy studies, with a spatial resolution of around 250 nm, have demonstrated how ageing affects overall brain vascular structure[5,15]. However, the BBB is composed of many components that are below 250 nm in size, e.g., tight junctions, basement membrane, or astrocyte endfeet. Thus, higher resolution microscopy is needed to quantify changes in these components. By contrast, electron microscopy (EM) can achieve sub-nanometre resolution, making it the preferred method by which to study the impact of age, injury and disease at the BBB ultrastructural level. Indeed, EM analyses have revealed several such changes in the vasculature of the ageing and diseased brains of both rodents and humans[8,14,16-18]. The downside of using EM is that analysis requires significant manual work which is highly time-consuming, leading to the unavoidable selection of very specific measurements to quantify, limiting the scope of the studies to a reduced number of structures, biological replicates, or brain areas. Furthermore, this type of analysis is susceptible to personal bias, as interpretation may be influenced by pre-existing hypotheses held by the researcher, leading to the skewed selection of evidence. As such, there is a

[1]Imperial College London, London, UK. [2]UK Dementia Research Institute, Care Research and Technology Centre, London, UK. [3]Centre for Discovery Brain Sciences, The University of Edinburgh, Edinburgh, UK. [4]UK Dementia Research Institute, The University of Edinburgh, Edinburgh, UK. [5]British Heart Foundation and UK Dementia Research Institute Centre for Vascular Dementia Research, London, UK. [6]University College London, London, UK. [7]Great Ormond Street Hospital NHS Foundation Trust, London, UK. [8]Present address: University College London, London, UK. [9]These authors contributed equally: Blanca Díaz Castro, Payam Barnaghi. ✉e-mail: nan.fletcher-lloyd17@imperial.ac.uk

need for an efficient method that can minimise subjectivity and provide a more rigorous and unbiased analysis.

The advent of deep learning techniques, particularly Deep Convolution Neural Networks (DCNNs), in the field of machine learning has revolutionised the automation of cell image analysis. Unlike traditional machine learning, these models use multi-layer hierarchical feature extraction, making them more suitable for processing complex morphology for classification, detection, and segmentation[19]. In the last few years, several studies have leveraged deep learning techniques to study brain cells. Suleymanova et al. applied a DCNN-based method to count the number of astrocytes in different brain regions of rats in immunohistological images[20]. Kayasandik et al. took this work one step further, using multiscale directional filters for improved astrocyte detection in fluorescent microscopy images of mouse brains[21]. Most recently, Heckenbach et al. presented deep learning models that could predict cellular senescence based on nuclear morphology. Nuclei were detected using the image segmentation network U-Net, before a predictor was used to generate senescence scores in human fibroblasts[22]. They also showed that their model could identify senescence in murine astrocytes among other cell types. Deep learning has also been used to study BBB function, primarily focusing on improving the predictability of its permeability to support the discovery of drug compounds able to cross the BBB[23–28]. Importantly, however, no studies have yet applied deep learning techniques to study the BBB at a structural level. Inspired by these works, our goal was to develop a deep learning model that could identify BBB electron microscopy images of aged mouse brains from young mouse brains based on the BBB architecture, which includes morphological, structural and textural features.

This study presents 3BTRON: A Blood-Brain Barrier Recognition Network for analysing BBB architecture in electron microscopy images, with a focus on capillaries, which make up approximately 80% of the blood vessels in the brain. As a proof-of-concept, we demonstrate the utility of 3BTRON applied to the identification of the BBB architecture of the brains of aged mice, establishing its potential as an investigative tool. Through optimisation, we establish stratification thresholds for age likelihood to enhance the network's flexibility for contextual translation, before using feature importance to reveal the parts of each image that contribute most towards the age likelihood score predicted for that image. The proposed approach has been evaluated on a dataset of 359 images taken from 23 wild-type mice (Fig. 1 presents an overview).

Our work offers an automated framework for analysing BBB architecture in different brain regions. The use of high-resolution image data combined with deep transfer learning results in a high-throughput pipeline that can be used to support research into the BBB with increased efficiency. By providing a standardised, data-driven assessment of the BBB

architecture, this framework ensures greater reliability and generalisability of findings. In the context of ageing, our model can be used as a tool to investigate researcher hypotheses, for example, by guiding the discovery of age-related changes to the BBB architecture that may lead to the brain's regional vulnerability to neurodegenerative diseases, or to efficiently assess the effectivity of treatments targeting vascular ageing at the BBB structural level.

## Results
### Model performance
In total, our dataset was comprised of 359 images, collected from 23 mice (11 female, 12 male). Of these images, 161 are from aged mice (mean age in days of 577, ± 40 to the nearest whole day) and 198 are from young mice (mean age in days of 98, ± 24 to the nearest whole day). Images were taken across three different brain regions: the corpus callosum (labelled CC), the hippocampus (labelled HC), and the prefrontal cortex (labelled PFC). Figure 1 shows a breakdown of the image data composition by demographics (age and sex) and anatomy (brain region), as well as a more detailed breakdown of sample distribution by mouse.

We used transfer learning and data augmentation techniques, resulting in over 1000 samples per class, to train a model capable of distinguishing between the BBB architecture of aged and young mouse brains from high-resolution electron microscopy data. We trained the model using a binary classification task based on the chronological age of the mice recorded at the time of culling. We evaluated and compared the effectiveness of our model using four different architectures to extract image features: Resnet50, MobileNetv2, VGG16, and VGG19. Table 1 shows the results of the 10-fold cross-validation on all the models tested.

We found the best-performing classification model was ResNet50 with L2 regularisation. Training on an NVIDIA Ampere A100 GPU (see the Methods for a more detailed specification), over 11 epochs, the model processed 28,512 images in 34 s at a rate of 52.4 (3.s.f.) batches per second (average time per epoch of 3.09 s (3.s.f.), average time per batch of size 16 of 19.1 (3.s.f.) ms), which is equivalent to 1.19 (3.s.f.) ms per image on average. Table 2 presents this model performance on both the validation and test data (definitions of evaluation metrics can be found in Supplementary Information Section 2).

### Stratification of age likelihood
We calculated stratified age likelihood scores as discussed in Methods: Stratification of probability estimates for reporting model uncertainty. These scores are a measure of how confident the model is in its predictions. By varying the stratification thresholds, we were able to balance the sensitivity and specificity for the 'Green' (young) and 'Red' (aged) groups, while

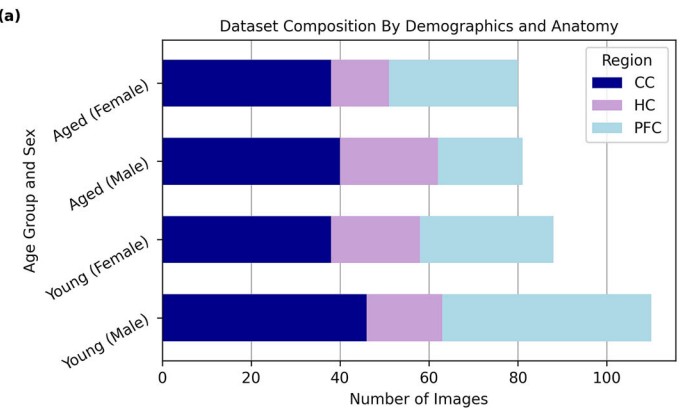

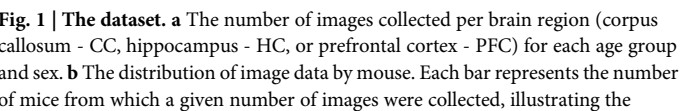

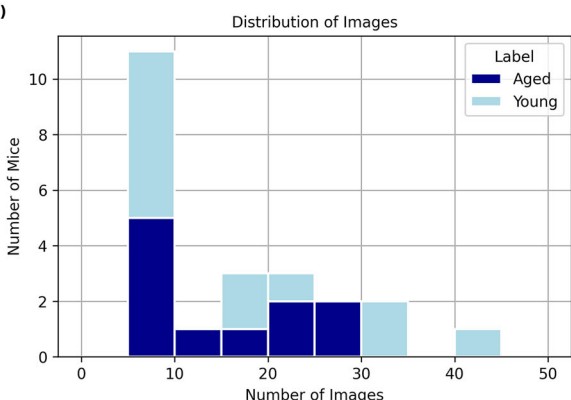

**Fig. 1 | The dataset. a** The number of images collected per brain region (corpus callosum - CC, hippocampus - HC, or prefrontal cortex - PFC) for each age group and sex. **b** The distribution of image data by mouse. Each bar represents the number of mice from which a given number of images were collected, illustrating the variability in image counts across mice. Image counts are grouped into bins of 5 images. Colored bars represent the two different experimental groups; aged (dark blue) and young (light blue). We see that the vast majority of images have been collected from different mice, which will reduce bias from overfitting in the model.

**Table 1 | Mean (95% CI) % of sensitivity, specificity, and area under the precision-recall curve of all prediction models across 10-fold cross-validation**

|  | Sensitivity | Specificity | AUC Precision-Recall |
|---|---|---|---|
| **ResNet50** | 67.6 (65.2–69.9) | 72.6 (69.4–75.8) | 76.7 (73.8–79.7) |
| MobileNet | 69.4 (64.2–74.6) | 64.2 (60.0–68.5) | 72.8 (70.2–75.4) |
| VGG16 | 54.5 (49.7–59.2) | 76.7 (72.6–80.7) | 71.8 (68.5–75.1) |
| VGG19 | 60.3 (53.9–66.7) | 75.5 (70.7–80.3) | 73.7 (70.1–77.3) |

Bold indicates the best performing model.

**Table 2 | Mean (95% CI) % of sensitivity, specificity, and area under the precision-recall curve of the prediction model on the different data splits**

| Before Stratification |  |  |  |
|---|---|---|---|
|  | Sensitivity | Specificity | AUC Precision-Recall |
| **Validation** | 67.6 (65.2–69.9) | 72.6 (69.4–75.8) | 76.7 (73.8–79.7) |
| **Test** | 68.8 | 52.6 | 63.9 |
| **After Stratification** |  |  |  |
|  | Sensitivity | Specificity | Precision |
| **Validation** | 79.0 (75.5–82.5) | 79.5 (75.3–83.6) | 76.7 (72.2–81.2) |
| **Test** | 77.8 | 80.0 | 77.8 |

These results are reported before and after stratification is performed.

limiting the number of samples about which the model could not confidently predict. Here, we select thresholds [0%, 25%], [25%, 75%], and [75%, 100%] (following interval notation) for the 'Green', 'Amber', and 'Red' groups, respectively. Table 2 shows the model performance after stratification when grouping the predictions on the 'Green' and 'Red' groups. Figure 2 reports the distribution of the model's predictions by group (including the 'Amber' group) on the test set. We also report the difference in mean predicted probabilities between images that the model has classified as young and aged (after stratification) for each sex and brain region (see Tables 3 and 4). Analysis of model reliability and calibration can be seen in Supplementary Information Figure S3 in Supplementary Information Section 5. Additionally, in Supplementary Information Section 7, we compare model performance across tissues from different brain regions and male and female mice (see Supplementary Information Tables S1 and S2).

### Feature importance

The most important regions influencing predictions according to the convolutional layers were identified using Gradient-weighted Class Activation Mapping (Grad-CAM), a technique used in deep learning for producing explainable predictions and calculating contributions from the spatial features of images[29]. In early layers, convolutional neural networks learn to detect low-level features such as edges and textures. In deeper layers, these networks capture higher-order patterns such as objects and spatial arrangements. Here, we show Grad-CAM visualisations generated from the feature maps of two different convolutional layers (the final convolutional layers of block 4 in stage 2 and block 3 in stage 4, respectively) of the neural network, with warm colours (red or yellow) showing regions of the image to which the model paid the most attention and cool colours (blue or green) indicating areas that were less important to the final decision. This approach allowed us to better understand the prediction made for each image and the contributions to the age likelihood score. Figure 3 shows this breakdown on three images from the test set based on their post-stratification grouping: a correctly predicted certain positive (aged) 'red' example, a correctly predicted certain negative (young) 'green' example, and an example that the model could not confidently predict ('amber'). We also present Grad-CAM visualisations of images that the model confidently predicted incorrectly (see Fig. 4).

### Model behaviour on the middle-age brain

To further assess the model's ability to distinguish between the BBB architecture of aged and young mouse brains, we applied the final trained model to an unlabelled dataset of 52 images taken from the corpus callosum and prefrontal cortex of the brains of 3 middle-age (366 days) female mice, an age group not explicitly present in the binary classification task used for model training. In Table 5, we report the percentage of images predicted as young and aged for each brain region, as well as the percentage of images about which the model was uncertain. Figure 5 shows the breakdown of the model's predictions on three images using Grad-CAM heatmaps: an image predicted positive (aged), an image predicted negative (young), and an image that the model could not confidently predict.

### Discussion

The BBB plays a crucial role in maintaining brain health and function. Indeed, the breakdown of the BBB has been recognised as an early biomarker for human cognitive dysfunction[30]. Therefore, to better understand how the ageing process increases vulnerability to disease, it is essential to unveil the impact of ageing on the architecture of the BBB. Moreover, given the different susceptibilities of brain regions to various types of pathology, it is important to explore regional differences during ageing. The primary aim of this study was to develop a model that reflects the inherent complexity of this field of study, rather than to maximise predictive performance through overengineering. Our framework is designed to be used as a scientific tool to guide and accelerate discovery, offering insights and supporting hypothesis generation aligned with domain understanding. As such, unlike models built for automated decision-making that focus on performance optimisation by tuning for metrics such as sensitivity, specificity and precision, our approach consciously avoids overfitting to these criteria (which tends to be done at the expense of both generalisability and transparency), instead focusing on constructing a system that balances usability and interpretability, ensuring our model remains a practical aid in exploratory research and knowledge generation, as opposed to a black-box solution for complete decision automation.

We present a deep transfer learning pipeline for estimating age likelihood using electron microscopy image data, combined with demographic and anatomical data. We considered several deep learning models for image feature extraction and found that ResNet50 attained the top performance (sensitivity of 68.8% and specificity of 52.6%). The ratios of positives (images from aged mice) to negatives (images from young mice) in the test set is 1 : 1.25. Through stratification, age likelihood scores were transformed into three tuneable groups allowing us flexibility in the management of model accuracy and fairness. Following this, the performance on the 'Green' and 'Red' groups was significantly improved, achieving a sensitivity of 77.8%, and a specificity of 80.0% (see Supplementary Information Section 2 for definitions of evaluation metrics). A breakdown of these predictions as seen in Fig. 2 and Tables S1 and S2 in Supplementary Information Section 7 showed that re-grouping the model's predictions based on the age likelihood score ensured similar levels of accuracy were achieved in each sex and across all three brain regions. While these stratification thresholds increased model uncertainty, we chose to prioritise correct certain predictions for the purposes of this research scenario. When we introduced a set of images taken from middle-age mouse brains, the model classified the majority of images as either aged or returned an uncertain prediction. These findings suggest that the model has learned a meaningful age-related decision boundary by which it can reasonably separate young and aged mouse brains. These results also suggest that the model was able to identify more subtle age differences. Considering brain region, we see that just over half of the images from the corpus callosum of middle-age mice were classified as aged, while the model returned uncertain predictions about the majority of the images from the prefrontal cortex (see Table 5). These results indicate that the corpus callosum of middle-age female mouse brains are more similar to aged mouse brains than young. Overall, the application of our framework to middle-age mouse brains emphasises the potential of our model to help reveal the timeline of region-specific age-related changes.

**Fig. 2 | Final model stratification breakdown.**
Analysis of stratified groupings produced by evaluating the final model on the held-out test set using optimised thresholds. `Green' represents the samples that the model confidently predicts are from a young mouse, `Amber' the samples that the model is not confidently predicting, and `Red' the samples that the model confidently predicts are from an aged mouse. Each group is further broken down by the true age of the samples, then by the sex of the mouse (female - F/male - M), and finally by brain region (corpus callosum - CC, hippocampus - HC, or prefrontal cortex - PFC).

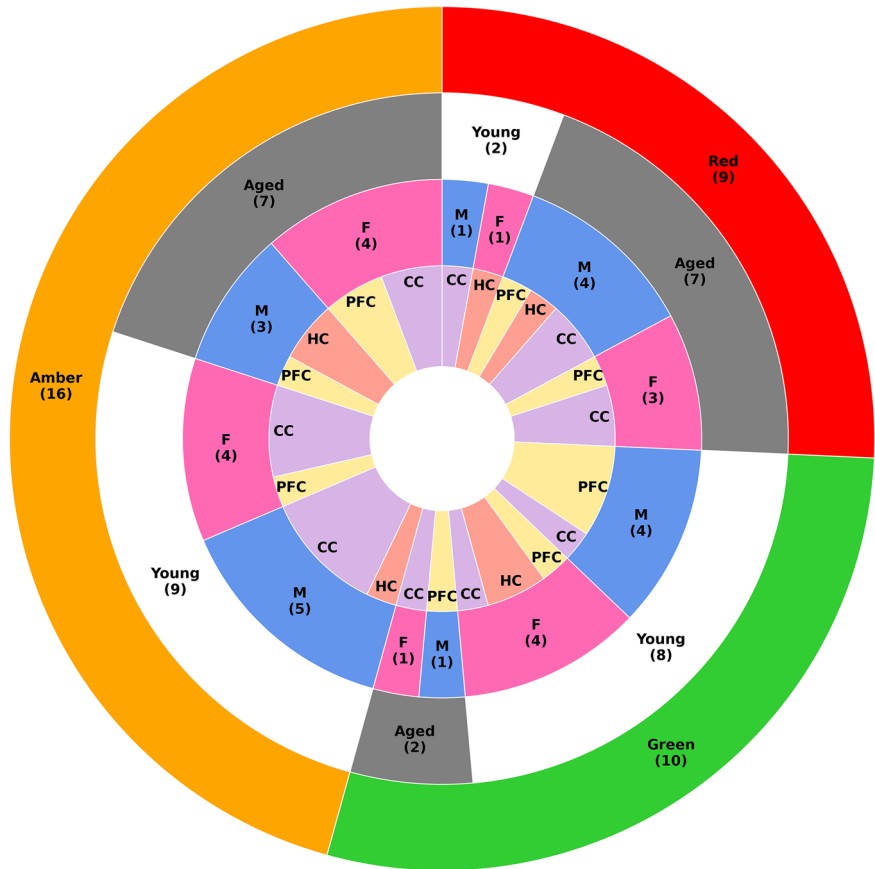

**Table 3 | Difference in mean predicted probability for images classified as young and aged by the model for each sex after stratification**

| Sex | Aged | Young | Difference |
|---|---|---|---|
| Female | 88.5 | 17.3 | 71.3 |
| Male | 90.1 | 15.6 | 74.5 |

**Table 4 | Difference in mean predicted probability for images classified as young and aged by the model for each brain region (corpus callosum - CC, hippocampus - HC, or prefrontal cortex - PFC) after stratification**

| Brain Region | Aged | Young | Difference |
|---|---|---|---|
| CC | 92.6 | 22.7 | 69.9 |
| HC | 83.4 | 16.7 | 66.6 |
| PFC | 87.4 | 12.5 | 74.9 |

Our approach uses Grad-CAM analysis to highlight the spatial features that most strongly contributed to the age likelihood score for a given image. Here, we chose to present the heatmaps from two different convolutional layers of the model (one from an early layer of the model and one from the final convolutional layer of the model, respectively) to illustrate the spatial features important in the early and final stages of classification; however, the advantage of using Grad-CAM is that it can be applied to any convolutional layer in a neural network, which enables us to create a comparative set of maps that can provide a sense of the hierarchical feature extraction of the model from local to contextual, maximising the utility of our deep learning framework to provide the most comprehensive view of the features determining the model's decision making and enabling us to derive the most

meaningful conclusions. Expert analysis of the Grad-CAM images displayed here show that regions containing or adjacent to the capillary, including the astrocyte endfeet, were important in the early stages of classification. This may relate to the observation that astrocyte endfeet in rodents grow larger with age[14]. For the final stages of classification, regions of the images containing the brain parenchyma were also important, with the model likely recognising spatially linked changes as an indirect signature of the ageing BBB, for example, potential morphological distortions that could be due to alterations in the volume of extracellular space.

Finally, as an additional functionality of this framework, we generated a score to better understand the findings of the model with respect to age for each sex and brain region. Our analysis shows that the model finds the greatest difference between young and aged BBB architecture in the prefrontal cortex (74.9%) and male mice (74.5%) (see Tables 3 and 4), respectively, which is in line with the literature published on both rodent and human brains[17,31].

This study contains several strengths, as well as a few limitations that would also serve as focuses for future research. The use of deep learning techniques allows us to detect differences that might be difficult to observe on manual inspection, while ensuring greater objectivity in the findings. Automated analysis also greatly increases the number of images that can be analysed in a given time period. Furthermore, by harnessing the power of transfer learning, we were able to apply the knowledge obtained from training deep models on millions of natural images to our electron microscopy data, allowing us to achieve good performance with a comparatively small number of images ($n = 359$). We compared four well-established architectures representing a balanced set of state-of-the-art (ResNet50), traditional (VGG16 and VGG19), and efficient (MobileNetv2) models for image classification, with all four models equipped to handle diverse image types having been trained on the ImageNet database[32–35]. The final model used the ResNet50 architecture, a powerful deep network for learning complex, hierarchical features. This model provides a good first indication

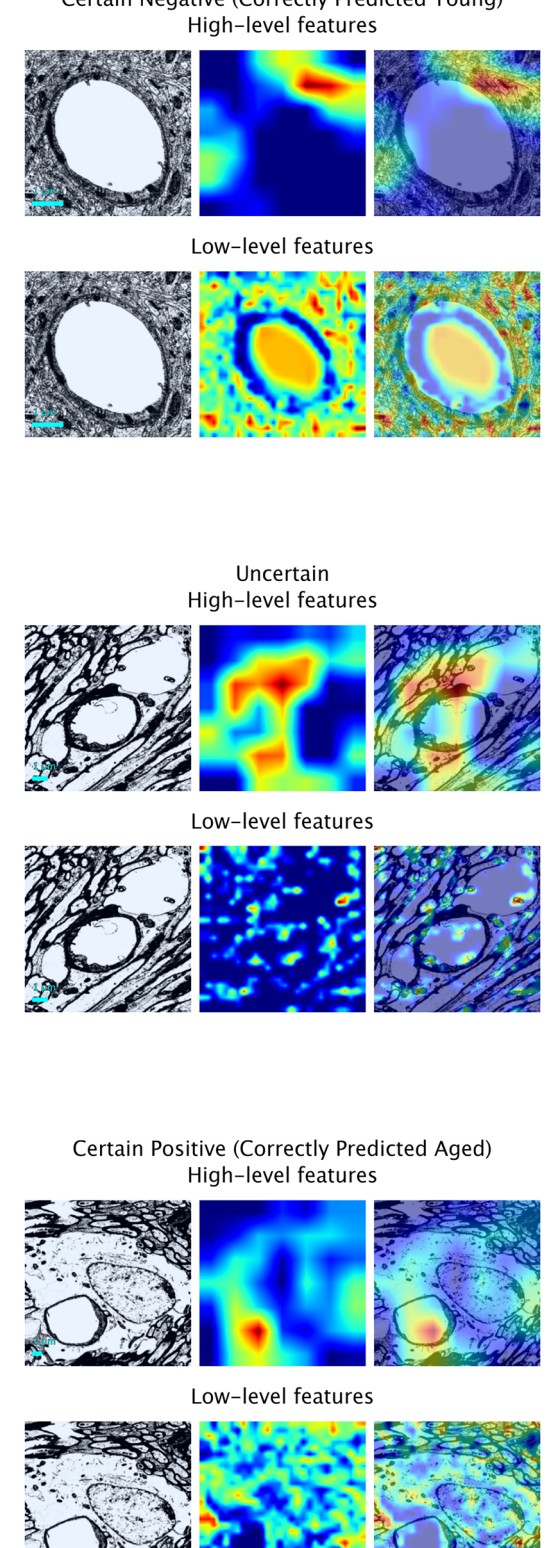

**Fig. 3 | Visualisations of contributing spatial features for early-stage and final classification of images in the test-set based on their post-stratification grouping.** Correctly predicted examples are shown for the 'red' and 'green' groups. Overlaid Gradient-weighted Class Activation Mapping heatmaps show how strongly different regions of each input image contribute to the specific class prediction given to that image by the final model. Warm colours (red or yellow) show regions to which the model has paid the most attention and cool colours (blue or green) indicate areas that were less important to the final decision. Scale bar: 1 $\mu$m.

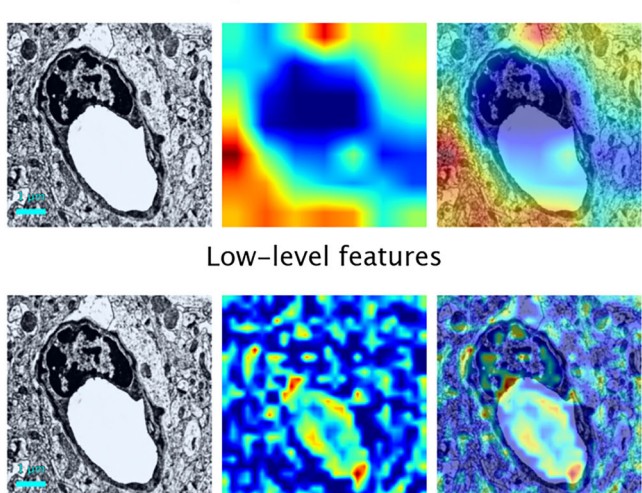

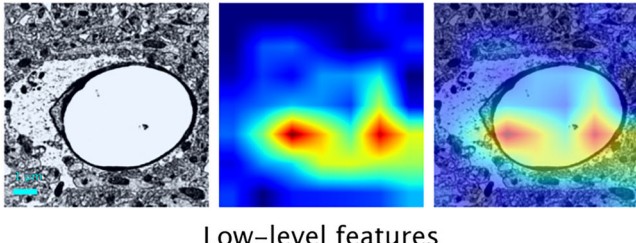

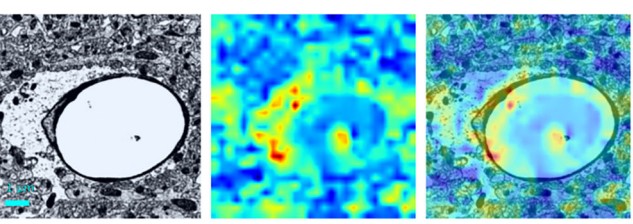

**Fig. 4 | Visualisations of contributing spatial features for early-stage and final classification of images in the test set that the model confidently predicted incorrectly.** Overlaid Gradient-weighted Class Activation Mapping heatmaps show how strongly different regions of each input image contribute to the specific class prediction given to that image by the final model. Warm colours (red or yellow) show regions to which the model has paid the most attention and cool colours (blue or green) indicate areas that were less important to the final decision. Scale bar: 1 $\mu$m.

as to the differences in BBB architecture in young and aged mice; however, the required image input size of all four baseline models is 224-by-224 pixels. At this resolution, the model might struggle to capture fine-grained details unless they are large or distinct enough within the image. More domain-specific models (such as DenseNet121 and U-Net) might provide greater insight into age-related differences at a finer granularity[36,37]. However, these models are not typically designed for image classification and can introduce unnecessary complexity leading to overfitting. In addition, the use of such models is computationally expensive. As a key focus of this work has been to create a framework that can be used by experimentalists who may not have access to extensive computational resources, future directions would

**Table 5 | Percentage of images predicted young and aged for each brain region (corpus callosum - CC and prefrontal cortex - PFC), as well as the percentage of images about which the model was uncertain for images from middle-age mouse brains**

| Brain Region | Total Images | Aged | Young | Uncertain |
|---|---|---|---|---|
| CC | 29 | 55.2 | 13.8 | 31.0 |
| PFC | 23 | 26.1 | 8.7 | 65.2 |

consider ensemble architectures that can leverage the generalisability and comparative computational efficiency of ResNet50 (our model processed each image in just over 1 ms) combined with the additional complexity of more specialised models. An alternative we considered but did not select was to pre-train the model on electron microscopy data. There are a limited number of large-scale, high-quality electron microscopy datasets available. To the best of our knowledge, ours is the first publicly available electron microscopy dataset collected for the purpose of investigating the changes that occur in the blood-brain barrier during the ageing process. As electron microscopy datasets are generally biased towards certain tissue types, anatomical regions or cellular structures, and are often acquired under different acquisition settings, models pre-trained on these datasets risk overfitting to the characteristics of these specific datasets - such as contrast levels, resolution, or structural composition - thereby reducing their ability to generalise to different electron microscopy datasets. As the main objective of this work was to model the real-world heterogeneity of this data, we wanted to avoid pre-training on narrow defined datasets to ensure that the system remains robust across a diverse range of imaging conditions and biological contexts. By using ImageNet for pre-training, we were able to leverage a set of general features that are transferable even in domains like electron microscopy. Training a model on diverse data, such as ImageNet data, can actually improve robustness and make it easier to analyse errors that might reflect meaningful uncertainty.

In Supplementary Information Section 5, we discuss the model reliability and calibration and find that our model overestimates age likelihood, likely because of an imbalance in the data across brain regions. This motivates the applied age likelihood stratification which allows us to balance sensitivity and specificity with model uncertainty; however, in different research scenarios the trade-off between sensitivity and specificity should be re-evaluated to ensure the groups remain well calibrated for the task at hand. This work has focused on identifying age-associated changes in the architecture of the BBB in three different brain regions (see Supplementary Information Table S2 in Supplementary Information Section 7 for a preliminary analysis of model performance across each brain region). Since our model is primarily designed for classification, it will first prioritise global patterns that are consistent across the whole dataset; however, by providing the model with anatomical information, we are providing the model with an explicit signal to treat different brain regions separately, so it may also learn to prioritise region-specific features that are important for the task of classification. Further work could look to apply our approach to datasets of electron microscopy images taken from single brain regions to improve our understanding of region-specific changes in BBB architecture. While we recognise that alternative integration strategies might improve performance, as the primary goal of this work was to develop a usable framework for researchers without deep learning expertise, simple concatenation is easier to implement and apply consistently. Furthermore, the use of simple concatenation facilitates the tracing of the effect of different categorical variables (such as sex and brain region), thereby improving interpretability and enabling the model to be audited for fairness.

Our feasibility study was conducted using electron microscopy images of brain tissue samples taken from wild-type murine models. As this data is collected as part of an ongoing animal study in the Díaz Castro lab, we also have an ethical obligation to ensure that animal sacrifice is conducted responsibly, limiting the data available to us. While larger studies are

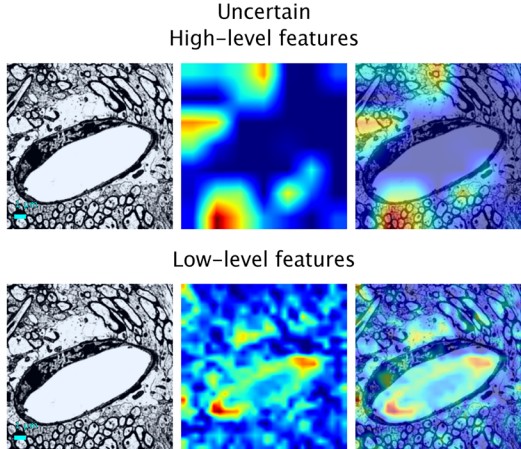

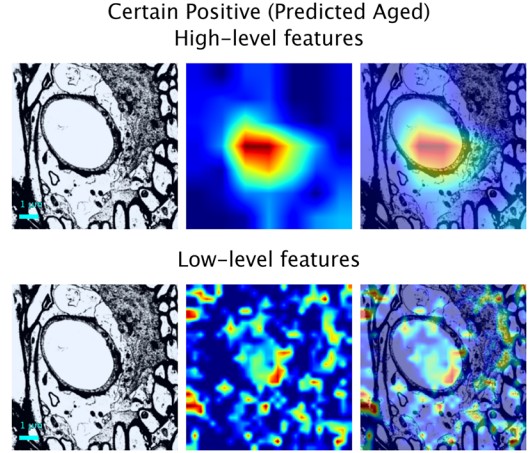

**Fig. 5 | Visualisations of contributing spatial features for early-stage and final classification of images from middle-age mice.** Overlaid Gradient-weighted Class Activation Mapping heatmaps show how strongly different regions of each input image contribute to the specific class prediction given to that image by the final model. Warm colours (red or yellow) show regions to which the model has paid the most attention and cool colours (blue or green) areas that were less important to the final decision. Scale bar: 1 $\mu$m.

required, for example, to investigate the impact of incrementally unfreezing layers of the model to optimise configuration for transfer learning (not implemented here due to the risk of introducing instability or overfitting to the training data), by combining transfer learning with data augmentation techniques, we improve the generalisability power of our framework. Moreover, despite commonalities, there are a number of important age-related differences between mice and humans[38]. Additional future work will investigate the utility of our model applied to human brains, as well inter-species differences in BBB architecture and the changes experienced during the ageing process.

Here, we provide preliminary evidence for an approach to identify the BBB architecture of aged brains which, with further testing, has the potential to accelerate research in the field of ageing and disease. To the best of our knowledge, this is the first such study to be conducted in this area, making this the largest currently available dataset. The proposed model takes advantage of the advancements made in deep transfer learning to create a scalable and efficient framework that can be easily applied to a variety of datasets. Moreover, by applying stratification to our age likelihood scores, this allows us to modify group thresholds for different datasets and calibrate the model for improved performance in different research scenarios. Finally, Grad-CAM analysis (such as in Figs. 3 and 5) enables the presentation of explainable results, allowing researchers to understand which features most strongly contribute to the algorithm's final decision and providing the opportunity for new hypothesis generation that could be tested in the future. For instance, one could hypothesise that our model is sensitive to features related to endfoot swelling. Then one could measure if the images classified by the model as aged display a higher rate of swollen endfeet.

3BTRON is a deep transfer learning framework developed to enhance the analysis of BBB architecture in electron microscopy images. As an investigative tool, 3BTRON can be very powerful, offering researchers the possibility of more targeted manual analyses, focused on features highlighted by the model, such as the brain regions most susceptible to age-related changes, or the most relevant aspects of an image with respect to the class prediction for that image. The use of 3BTRON has the potential to help uncover key insights into the impact of ageing and neurodegenerative disease on the architecture of the BBB, accelerating research in biomarker and drug discovery for age-related diseases. Whilst this work focused on the BBB, this framework can be adapted to be applied to the investigation of other cell types and brain structures.

## Methods

### Study design and ethics
Electron microscopy images were obtained from C57Bl/6J mice as part of an ongoing study in the Díaz Castro lab. The mice were sourced from Charles River Laboratories. Animal experiments were conducted in accordance with the national and institutional guidelines ([Scientific Procedures Act] 1986 (UK), and the Council Directive 2010/63EU of the European Parliament and the Council of 22 September 2010 on the protection of animals used for scientific purposes) and had a full Home Office ethical approval. We have complied with all relevant ethical regulations for animal use. Mice were housed with no restrictions to food and water in a 12-hours light/dark cycle. The age of the mice was recorded at the time they were culled. A lethal dose of the anaesthetic sodium pentobarbital (700 mg/kg) was administered intraperitoneally.

### Electron microscopy
Mice were perfused with PBS followed by fixative solution (4% paraformaldehyde w/v and 2% glutaraldehyde v/v in 0.1 M phosphate buffer). The brains were extracted, post-fixed in the same fixative solution for 24 h at 4 °C and sectioned into 50 $\mu$m thickness sections using a vibratome. Sections were post-fixed with 1% osmium tetroxide in 0.1 M phosphate buffer for 30 min, then dehydrated in ascending concentrations of ethanol dilutions (50%, 70%, 95%, and 100%) and acetone. Sections were then immersed in resin and left to set overnight at room temperature. The sections were subsequently placed on microscope slides, covered with coverslips, and the resin was cured at 65 °C for three days. The dissected areas were cut from the slides, mounted on plastic blocks, and ultrathin sectioned (60 nm thick) for electron microscopy. Each section was contrasted with lead citrate and uranyl acetate and imaged using a JOEL TEM-1400 plus electron microscope. Magnifications were selected to visualise each capillary along with the surrounding perivascular region for assessment. The resulting images are RGB (used for structural distinction) high-resolution (nanometre scale) electron microscopy images depicting the ultrastructural organisation of the blood-brain barrier, showing a cross-sectional view of a brain capillary, clearly visualising the endothelial cells forming the capillary wall, the highly specialised tight junctions, and the supporting mural cells, astrocyte endfeet, and basement membrane between them.

### Data characteristics and pre-processing procedures
In addition to image data, we also included sex and brain region data to the inference layer to allow for research interpretability and improve model performance and generalisability. To make the input data compatible with our model, categorical data was transformed into a numeric format using one-hot encoding to avoid ordinal misinterpretation. To prepare the image data for our models, we performed a series of initial image-pre-processing steps to improve generalisability. We first used histogram equalisation to enhance contrast to make it easier for our models to distinguish key features. We then performed image thresholding to reduce noise, simplifying the image to further improve pattern recognition. Following this, standard pre-processing steps were used to ensure the image data matched the expectations of the pre-trained architectures tested during model development. The images were resized from a resolution of 4096 x 4096 pixels to 224 x 224 pixels (as is necessary for the pre-trained models being evaluated), converted to an RGB format, and pixel values scaled to the range of [−1, 1], before being converted to a tensor.

### Analysis platform
All analyses were performed on a secure computing environment at Imperial College London using Python version 3.8.10. Model training was performed on an NVIDIA Ampere A100 GPU, which features 6912, CUDA cores, 432 third-gen Tensor cores, 80 GB of memory with up to 2.0 TB bandwith. The Pandas[39] (version 2.2.1), NumPy[40] (version 1.26.4), Scikit-Learn[41] (version 1.4.2), and PyTorch[42] (version 2.2.2) packages formed much of our pipeline. All visualisations were plotted using Matplotlib[43] (version 3.8.4) and Seaborn[44] version (0.13.2).

### Methodology
To ensure generalisability, the dataset was split into training and testing subsets in a 90:10 ratio, stratified by age. The training data represented 90% of the total data ($n = 324$ samples) and the testing data represented the remaining 10% ($n = 35$ samples). A small number of images were of the same vessel taken using different magnifications, positioning or contrast. As such, one image was dropped from the test set after data pre-processing to avoid data leakage. During model development and whilst optimising model hyperparameters and thresholds, validation sets were produced by splitting the training data using stratified k-fold cross-validation following the application of data shuffling. See Supplementary Information Figure S2 in Supplementary Information Section 3 for a visualisation of this evaluation. Stratification ensured that each fold contained a balanced representation of each age group compared to the whole training dataset, which is particularly important here due to the slight imbalance in the age group distributions, and data shuffling prevented ordering effects from influencing the model's performance. Following each split, the training set underwent augmentation using a variety of techniques. To leverage the advantages of offline data augmentation without the storage overhead, we applied online-offline hybrid data augmentation to the training data before training the model. In each cross-validation split, the original training data is loaded 8 separate times, each time with a unique series of transformations. The final training data is a concatenated set of the original image data, alongside image sets generated by horizontal flipping, and rotation by 90°, 180° and

270°), respectively. For both hyperparameter tuning and threshold optimisation, this approach allowed us to reduce computational burden during training while augmenting the data to over 1000 samples per class. Finally, we used sensitivity, specificity and the area under the precision-recall curve to measure model performance (for definitions of metrics, please see Supplementary Information Section 2).

## Model development

We use transfer learning and combine image and categorical data to predict whether an image comes from the brain tissue of a young or aged mouse based on the BBB architecture. Pre-trained models initialised with pre-trained weights are used for feature extraction. These image features are then concatenated with the categorical data to create a single unified vector which is then passed through a fully connected neural network to make the final prediction. For an in-depth illustration of the network architecture see Supplementary Information Figure S1 in Supplementary Information Section 1. We tested four commonly used deep learning models for image classification: ResNet50, MobileNetv2, VGG16, and VGG19[32–35]. These particular models all share the same input image data expectations with respect to image size, number of channels in the image, and pre-processing normalization) as they were all pre-trained on ImageNet. We fine-tuned only the final layers of each architecture while keeping the earlier layers frozen to leverage the robustness of pre-trained representations and minimise the risk of the model learning noise or patterns specific to only the training data rather than generalisable features. Hyperparameter optimisation was performed using k-fold cross-validation (k = 10), with the model producing the highest mean precision-recall area under the curve score on validation data selected for the final analysis. See Supplementary Information Figure S2 in Supplementary Information Section 3 for a visualisation of this evaluation. Supplementary Information Section 4 contains information on the decisions made regarding each step of the model pipeline.

## Stratification of probability estimates for reporting model uncertainty

Using an approach previously developed in our group[45], probability estimates from the model are stratified into three groups (Green, Amber, and Red), which refer to low, medium, and high likelihood of the sample coming from an aged mouse, respectively. These groupings can be used as a measure of the model's uncertainty (how often the model is not confident in its predictions). By varying these thresholds, we can better balance levels of sensitivity and specificity with model uncertainty, allowing us to understand the key features that are used in the model's decision-making process for predicting the positive (aged) and negative (young) class, related to changes in the architecture of the BBB. This also allows our process to be flexible to different research scenarios and datasets.

Within this work, the optimal thresholds used in our final analysis were calculated using k-fold cross-validation (k = 5), using the model's predictions of the validation data after being re-trained with the best hyperparameters on the training data. See Supplementary Information Figure S2 in Supplementary Information Section 3 for a visualisation of this evaluation. We chose to use 5-fold cross-validation to prevent overfitting to the validation set by limiting the amount of model fine-tuning. This also reduced overhead, by balancing validation with computational efficiency. More information on stratification is included in Supplementary Information Section 6.

## Statistics and reproducibility

The dataset was comprised of 359 samples, which were split into training and testing subsets in a 90:10 ratio. The training data represented 90% of the total data (n = 324 samples) and the testing data represented the remaining 10% (n = 35 samples). We also applied the final trained model to an unlabelled dataset of 52 images from the brains of middle-age mice. A small number of images were of the same vessel taken using different magnifications, positioning or contrast. As such, one image was dropped from the test set after data pre-processing to avoid data leakage. The data presented in this study can be found on Zenodo at: https://doi.org/10.5281/zenodo.14845497. The final model, as well as all associated code, can be found on GitHub at https://github.com/tmi-lab/3BTRON. The code for experiments presented in this study will be made available by the corresponding author upon reasonable request. During model development and whilst optimising model hyperparameters and thresholds, validation sets were produced by splitting the training data (n = 324) using stratified k-fold (k = 10) cross-validation following the application of data shuffling. The final model was validated on a held-out test set (n = 35 samples). For the held-out test set, experimental group labels were removed prior to modelling.

## Reporting summary

Further information on research design is available in the Nature Portfolio Reporting Summary linked to this article.

## Data availability

The data presented in this study can be found on Zenodo at: https://doi.org/10.5281/zenodo.14845497[46].

## Code availability

To promote the sharing of resources, the final model, as well as all associated code, can be found on GitHub at https://github.com/tmi-lab/3BTRON[47]. The code for experiments presented in this study will be made available by the corresponding author upon reasonable request.

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

## Acknowledgements

This study is funded by the UK Dementia Research Institute [award number UK DRI-7002 and UK DRI-4007] through UK DRI Ltd, funded by the Medical Research Council (MRC), Alzheimer's Research UK, Alzheimer's Society and the UKRI Engineering and Physical Sciences Research Council (EPSRC) PROTECT Project (grant number: EP/W031892/1) and Alzheimer's Research UK (ARUK-NC2020-SCO). Katrine Gaasdal-Bech is a student on the Translational Neuroscience PhD Programme and is funded by Wellcome grant 218493/Z/19/Z. Infrastructure support for this research was provided by the NIHR Imperial Biomedical Research Centre (BRC), the UKRI Medical Research Council (MRC) and the funders were not involved in the study design, data collection, data analysis or writing the manuscript. Payam Barnaghi is also funded by the Great Ormond Street Hospital and the Royal Academy of Engineering. We would also like to acknowledge the contribution of Professor Anna Williams and Dr Jonathan Moss in the acquisition of the electron microscopy data.

## Author contributions

NFL: Conceptualisation, Methodology, Software, Formal analysis, Investigation, Data Processing, Writing—Original Draft, Review and Editing, Visualisation; IBF, KGB: Data Collection, Review and Editing; BDC: Conceptualisation, Review and Editing, Supervision, Funding Acquisition; and PB: Conceptualisation, Methodology, Review and Editing, Supervision, Funding Acquisition.

## Competing interests

The authors declare no competing interests.
