## [Transparent Peer Review file · Communications Biology]

3BTRON: A Blood-Brain Barrier Recognition Network

Corresponding Author: Ms Nan Fletcher-Lloyd

This manuscript has been previously submitted at another journal. This document only contains information relating to versions considered at Communications Biology.

Version 0:

Reviewer comments:

Reviewer #1

(Remarks to the Author)

Fletcher-Lloyd et al., present deep learning algorithms to examine vascular changes using the EM images. The EM provides sufficient resolution to examine detailed sub-cellular changes in vasculature upon aging. Deep learning algorithms can provide a good analytical solution to perform detailed feature detection in the images. Introduction provides good rationale why the EM analysis is needed to unveil changes in the BBB. However, the result sections is not well constructed to present convincing evidence of algorithms performance. Many components seem unclear and overall deliverables of the manuscript remain quite unclear.

I am evaluating this manuscript as a biologist who utilizes lots of deep learning tools for automatic feature detection in large 3D image datasets. I struggled to understand how the algorithms was trained, how the ground truth datasets was established, and what features are aimed to be detected.

These significantly dampens my enthusiasm on the manuscript.

- There is no explanation what distinct features help to identify difference between young and old.
It is not clear what was trained on, is there any ground truth data for different cellular compartments?

- How was ground truth data established?
Is it based on some human annotations?
If so, one or multiple persons independently annotating?

- There is no clear explanation of input image and what kind of features are detected by the algorithms.
Something like this should be clearly presented before the current Table to explain what algorithms are actually doing for given EM images.
Examples of false positive, false negative, true positive, etc would be helpful with the original input images.

- In Fig1b, it's unclear what x axis means?
Please either update the axis name or add more explicit explanation in the figure legend.

- Result section start with the performance of different deep learning architecture without even explaining how the algorithms were trained. Much methodological details can go to the method section but some explanation of approaches need to be included in the early section of results before performance metrics.

- Table 1 shows both sensitivity and specificity of algorithms is under 0.8.
I don't think this performance is impressive enough.
How good is good enough?
Has this algorithms performance compared to human annotation (ground truth)?

- Figure 3 showed some spatial features.
It's unclear what is what.
The manuscript claim that the algorithms can distinguish young from old brain images. But it's unclear what features are helping to distinguish between the two groups.

- It is unclear what the mean of the section with “the middle-age brain” with Table 4.
Is this showing that the algorithms are not doing well for the middle aged brains, or something else?

- There are multiple different vascular segments (e.g., arteriole, capillaries, venules).
It's unclear what type of vessels are included here.

- Considering huge heterogeneity of the vessel types, size variabilities, small field of view, etc, 359 images seem to be quite under-powered. Hence, this low sampling rate weakened general claim and usability of tools.

- What are the computational demands to run the algorithms?
How long it takes to do feature detection per image?
What kind of GPU was used, etc?

Reviewer #2

(Remarks to the Author)

Fletcher-Lyioyd et al. have introduced an analytical platform, 3BTRON, for evaluating the BBB architecture based on images extracted from EM. The authors employ transfer learning using pre-trained CNN architectures, including ResNet50, MobileNetv2, VGG16, and VGG19. This study demonstrates several strengths: 1) appropriate use of transfer learning given the relatively small dataset (359 images), 2) evaluation of multiple architectures, providing insights into their feature extraction capabilities, and 3) freezing of early layers while fine-tuning later layers- a relevant strategy for this domain. Combined with the novel conceptualization and use of EM-driven images, this study holds great potentials for advancing our understanding of BBB architecture, particularly in relation to fundamental biological variable such as age and sex. Moreover, the platform could serve as a valuable tool for investigating BBB remodeling during the progression of brain diseases (e.g., AD) or in response to pharmacological interventions.

However, several areas for improvement remain. First, the paper does not discuss different freezing strategies, such as varying the number of frozen layers. Implementing progressive unfreezing could help identify a more optimal configuration for transfer learning. Second, while multiple architectures were tested, there is no analysis of the feature representations they produced to elucidate why ResNet50 outperformed the others. Third, the authors do not address the domain shift between natural images (ImageNet) and electron microscopy (EM) images. This is likely a major factor contributing to suboptimal performance. Incorporating domain adaptation techniques—such as pre-training on EM-specific datasets—could enhance model generalizability and accuracy. Finally, while the use of simple concatenation to combine different modalities is common, it may not be the most effective fusion strategy. Exploring more advanced methods for multimodal integration could further improve performance and interpretability.

Version 1:

Reviewer comments:

Reviewer #1

(Remarks to the Author)

Thanks for addressing my comments.

I found that the revised manuscript is significantly improved with enhanced clarity and necessary technical details. Changes in the revised manuscript helped me to more clearly understand the underlying logic and the goal of the manuscript.

I have no further comments and I support its publication.

Reviewer #2

(Remarks to the Author)

The authors have thoroughly addressed my comments in the revised version of the manuscript.

Response to the review comments

We would like to thank the reviewers for their time and for their constructive feedback, which has helped us improve our work. We have revised the paper according to the reviewers' comments. All revisions are highlighted in the manuscript in blue. Please see a detailed point-by-point description of the changes below.

Reviewer 1

Comment 1, part a: *There is no explanation what distinct features help to identify difference between young and old. It is not clear what was trained on, is there any ground truth data for different cellular compartments?*

Comment 1, part b: *How was ground truth data established? Is it based on some human annotations? If so, one or multiple persons independently annotating?*

Thank you for your comment. We trained an end-to-end deep learning framework to train our model to distinguish between the blood-brain barrier architecture of aged and young mouse brains from electron microscopy images collected from an ongoing animal study being conducted in the Díaz Castro lab. We employed convolutional neural networks for automated feature extraction (features learned during training without explicit manual design), leveraging feature maps from the final convolutional layers of block 4 in stage 2 and block 3 in stage 4, respectively, to generate heatmaps highlighting the regions of each image that contributed most strongly to the model's prediction (Figures 3, 4 & 5). The model was trained on a binary classification task based on the chronological age of the mouse, as recorded at the time of culling. We have revised the Abstract, Methods, and Results to provide greater clarity on this (please see changes as highlighted in Abstract, lines 21 to 22, Methods, Study design and ethics, lines 341 to 342 and Results, Model performance, lines 118 to 120, respectively). Furthermore, in parts of the manuscript and supplementary material where we refer to the 'BBB architecture', (please see the Introduction, lines 85 to 86, and the Supplementary Material, line 686) we have rephrased the text to make it clear what we mean by the use of this term with respect to what the model is being trained on.

Comment 2: *There is no clear explanation of input image and what kind of features are detected by the algorithms. Something like this should be clearly presented before the current table to explain what algorithms are actually doing for given EM images. Examples of false positive, false negative, true positive, etc would be helpful with the original input images.*

Thank you for your comment. The input images are RGB (used for structural distinction) high-resolution (nanometre scale) electron microscopy images depicting the ultrastructural organisation of the blood-brain barrier, showing a cross-sectional view of a brain capillary, clearly visualising the endothelial cells forming the capillary wall, the highly specialised tight junctions, and the supporting mural cells, astrocyte endfeet, and basement that surround them. We have updated the Methods to provide a more detailed explanation of the input image as above (please see changes as highlighted in Methods, Electron microscopy, lines 360 to 365). We have also revised the Results to provide greater clarity on the types of features detected by our algorithm (please see the changes highlighted in the Results, Feature Importance, lines 151 to 156). Finally, as Communications Biology allows for up to 10 display items for article types, we have now included as additional figure presenting examples of a false positive and false negative that the model confidently predicted on our test set post-stratification (please see

newly added Figure 4 in Results, Feature importance, page 9). We have also revised the corresponding figure legend and in-text references (lines 160 to 164) to provide greater clarity on what is shown in each figure.

Figure 3, with revised subtitles.

Newly added Figure 4.

Comment 3: *In Fig1b, it is unclear what x axis means? Please either update the axis name or add more explicit explanation in the figure legend.*

Thank you for your comment. We have revised the figure legend to include a clearer explanation of the information presented in Figure 1b.

Please see changes as highlighted in the Introduction at the bottom of page 3.

Comment 4: *Results section start with the performance of different deep learning architecture without even explaining how the algorithms were trained. Much methodological details can go to the methods section, but some explanation of approaches needs to be included in the early section of results before performance metrics.*

Thank you for your comment. We originally followed the style and formatting guidelines of Communications Biology in structuring the paper. However, in response to your concern, we have now included a brief description of the dataset and a summary of the key methodology in the Results section (please see the changes highlighted in Model performance, lines 108 to 120

and 131 to 132) to improve understandability, and further details are provided in the Supplementary Material.

Comment 5: *Table 1 shows both sensitivity and specificity of algorithms is under 0.8. I don't think this performance is impressive enough. How good is good enough? Has this algorithms performance compared to human annotation (ground truth)?*

Thank you for your comment. In this work, we utilise a binary classification task to train our model to distinguish between the blood-brain barrier architecture of aged and young mouse brains using the chronological age of the mice recorded at the time of culling. The primary aim of this study was to develop a model that reflects the inherent complexity of this field of study, rather than to maximise predictive performance through overengineering. Our framework is designed to be used as a scientific tool to guide and accelerate discovery, offering insights and supporting hypothesis generation aligned with domain understanding. As such, unlike models built for automated decision-making that focus on performance optimisation by tuning for metrics such as sensitivity, specificity and precision, our approach consciously avoids overfitting to these criteria (which tends to be done at the expense of both generalisability and transparency), instead focusing on constructing a system that balances usability and interpretability, ensuring our model remains a practical aid in exploratory research and knowledge generation, as opposed to a black-box solution for complete decision automation. We have included this response as an additional statement addressing our decision in the Discussion (please see changes as highlighted in paragraph 1, lines 181 to 191).

Comment 6: *Figure 3 showed some spatial features. It's unclear what is what. The manuscript claim that the algorithms can distinguish young from old brain images. But it is unclear what features are helping to distinguish between the two groups.*

Thank you for your comment. We leveraged feature maps from two different stages (early and final) of our classification model to generate heatmaps highlighting the regions of each image that contributed most strongly to the model's prediction. Warm colours (red or yellow) indicate regions of the image to which the model paid the most attention, while cool colours (blue or green) denote areas that were less important to the final decision. We have revised the Discussion to provide a clearer and more detailed analysis of the Grad-CAM visualisation figures, supported by additional references (please see changes as highlighted in paragraph 3, lines 229 to 234 and paragraph 8, lines 325 to 328).

Comment 7: *It is unclear what the mean of the section with "the middle-age brain" with Table 4. Is this showing that the algorithms are not doing well for the middle aged brains, or something else?*

Thank you for your comment. This is correct. It is expected that the model will perform poorly on images taken from middle-aged mouse brains, as this is an ambiguous age category with respect to the binary classification task used for model training. As such, our findings suggest that the model has learned a meaningful age-related decision boundary. We have revised both the Results (please see changes as highlighted within Model behaviour on the middle-age brain, lines 165 to 174 and the legend for Figure 5) and Discussion (please see changes as highlighted in paragraph 2, lines 206 to 210 and 213 to 215) where relevant to better emphasise the reasoning behind this use-case and the importance of these results in the context of model performance.

Comment 8: *There are multiple different vascular segments (e.g., arteriole, capillaries, venules). It's unclear what type of vessels are included here.*

Thank you for your comment. We recognise that the term 'blood vessels' is too broad and that we need to be more specific. As such, we have revised the manuscript to make it clear that the focus of this work is on the capillaries of the brain.

Please see changes as highlighted in the following sections:

1. Introduction, lines 88 to 89.
2. Methods, Electron microscopy, within lines 360 to 365.
3. Discussion, paragraph 3, within lines 227 to 234.

Comment 9: *Considering huge heterogeneity of the vessel types, size variabilities, small field of view, etc, 359 images seem to be quite under-powered. Hence, this low sampling rate weakened general claim and usability of tools.*

Thank you for your comment. We agree that larger studies are required; however, to the best of our knowledge, this is the first study of its kind to be conducted in this area, making it the largest currently available dataset. Furthermore, as this data is collected as part of an ongoing animal study in the Díaz Castro lab, we have an ethical obligation to ensure animal sacrifice is conducted responsibly, limiting the data available to us. Nonetheless, aware of this potential limitation, we employed transfer learning (a technique that leverages knowledge obtained from deep models pre-trained on millions of natural images) combined with data augmentation, to enhance the generalisability of our framework. We have revised the paper to include information on the data augmentation process (originally presented in the Supplementary Material) in the Methods section (please see the changes highlighted in the Methodology, lines 402 to 411). We have also included a statement in the Discussion addressing data availability (please see changes as highlighted in paragraphs 7 and 7, lines 304 to 311 and 317 to 318, respectively).

Comment 10: *What are the computational demands to run the algorithms? How long does it take to run feature detection per image? What kind of GPU was used, etc?*

Thank you for your comment. We have revised the Methods (please see changes as highlighted in the Analysis platform, lines 382 to 384) to include a breakdown of the specifications of the GPU on which we trained our model. We have also revised the Results (please see changes as highlighted in Model performance, lines 124 to 127) to provide a detailed report on the model processing time. We also provide a summary of the processing time per image in the Discussion (please see the changes highlighted in paragraph 4, line 264).

Reviewer 2

Comment 1: *First, the paper does not discuss different freezing strategies, such as varying the number of frozen layers. Implementing progressive unfreezing could help identify a more optimal configuration for transfer learning.*

Thank you for your comment. Due to the limited volume of training data, we did not implement progressive unfreezing, as to do so risked introducing instability or overfitting. To mitigate this risk, we instead fine-tuned only the final layers of each architecture while keeping the earlier

layers frozen. In this way, we were able to leverage the robustness of pre-trained representations and minimise the risk of the model learning noise or patterns specific to only the training data rather than generalisable features. We have included an additional statement addressing this decision in the Methods (please see changes as highlighted in Model development, lines 425 to 428) and the Discussion (please see changes as highlighted in paragraph 7, lines 306 to 311).

Comment 2: *Second, while multiple architectures were tested, there is no analysis of the feature representations they produced to elucidate why ResNet50 outperformed the others.*

Thank you for your comment. We tested four deep learning models using a 10-fold cross-validation to tune hyperparameters. The model that produced the highest mean precision-recall area under the curve score on the validation data was selected for the final analysis. We originally presented these results as a table in the Supplementary Material. As Communications Biology allows for up to 10 display items for article types, we have now moved this to the Results section (please see the changes highlighted in [Model performance], line 122).

Comment 3: *Third, the authors do not address the domain shift between natural images (ImageNet) and electron microscopy (EM) images. This is likely a major factor contributing to suboptimal performance. Incorporating domain adaptation techniques—such as pre-training on EM-specific datasets—could enhance model generalizability and accuracy.*

Thank you for your comment. There are a limited number of large-scale, high-quality electron microscopy datasets available. To the best of our knowledge, ours is the first publicly available electron microscopy dataset collected for the purpose of investigating the changes that occur in the blood-brain barrier during the ageing process. As electron microscopy datasets are generally biased towards certain tissue types, anatomical regions or cellular structures, and are often acquired under different acquisition settings, models pre-trained on these datasets risk overfitting to the characteristics of these specific datasets – such as contrast levels, resolution, or structural composition – thereby reducing their ability to generalise to different electron microscopy datasets. As the main objective of this work was to model the real-world heterogeneity of this data, we wanted to avoid pre-training on narrow defined datasets to ensure that the system remains robust across a diverse range of imaging conditions and biological contexts. By using ImageNet for pre-training, we were able to leverage a set of general features that are transferable even in domains like electron microscopy. Training a model on diverse data, such as ImageNet data, can actually improve robustness and make it easier to analyse errors that might reflect meaningful uncertainty. We have included this response as an additional statement addressing our decision in the Discussion (please see changes as highlighted in paragraph 5, lines 265 to 281).

however, to the best of our knowledge, this is the first study of its kind to be conducted in this area, making it the largest currently available dataset.

Comment 4: *Finally, while the use of simple concatenation to combine different modalities is common, it may not be the most effective fusion strategy. Exploring more advanced methods for multimodal integration could further improve performance and interpretability.*

Thank you for your comment. While we recognise that alternative integration strategies might improve performance, as the primary goal of this work was to develop a usable framework for researchers without deep learning expertise, simple concatenation is easier to implement and apply consistently. Furthermore, the use of simple concatenation facilitates the tracing of the effect of different categorical variables (such as sex and brain region), thereby improving interpretability and enabling the model to be audited for fairness. We have included this response as an additional statement addressing our decision in the Discussion (please see the changes highlighted in paragraph 6, lines 297 to 302).